

# Fluvial boulder transport controls valley blocking by earthflows in the California Coast Range, USA.

Noah J. Finnegan[1], Kiara N. Broudy[1], Alexander L. Nereson[1], Joshua J. Roering[2], Alexander L. Handwerger[3], Georgina Bennett[4]

[1]Department of Earth and Planetary Sciences, UC Santa Cruz, Santa Cruz, CA, USA 95064, USA
[2]Department of Earth Sciences, University of Oregon, Eugene, OR, 97403-1272, USA
[3]Jet Propulsion Laboratory, California Institute of Technology, Pasadena, CA 91109, USA
[4]School of Environmental Sciences, University of East Anglia, Norwich, Norfolk, NR4 7TJ, UK

*Correspondence to*: Noah J. Finnegan (nfinnega@ucsc.edu)

**Abstract.** At two similar sites in California's Franciscan Mélange (Arroyo Hondo, central California and the Eel River, northern California), earthflows impinge on river channels with drainage areas that differ by a factor of 30. We compare these sites to explore how river flow depth, width, and velocity control river resilience to valley blocking and aggradation that occurs as earthflows deliver large quantities of coarse sediment to the channel network. We measure the size distribution of earthflow-derived boulders (>30 cm) delivered to each channel and use USGS stream gages to quantify the mobile fraction of boulders for a 2-year recurrence interval flood. For Arroyo Hondo, only the top ~10% of boulders are immobile (> 2.4 m diameter), however, this portion of the distribution represents ~80% of the volume of coarse, earthflow-derived material. For the Eel River, the top ~1% of boulders are immobile (> 4.9 m diameter), a fraction that represents only ~20% by volume. Satellite imagery shows that immobile boulders in Arroyo Hondo jam the entire channel and coincide with knickpoints and aggradation for km's upstream. By comparison, immobile boulders in the Eel River are sparsely distributed and confined to the edges of the channel. Moreover, the Eel River valley and long profile show little evidence of perturbation despite numerous well documented active earthflows along its length. This contrast suggests valley blocking is very sensitive to the mobility of the coarsest fraction of sediment. In this way, earthflow-impacted channels may act like step-pool channels, where channel-spanning boulder jams locally impede coarse sediment transport. In support of this view, the ratio of channel width to the threshold diameter for immobile boulders on Arroyo Hondo is ~5 (as opposed to ~25 for the Eel River), suggesting the river is susceptible to jamming. Our results imply that the lower drainage area, upper portions of earthflow-dominated catchments may be particularly prone to blocking. Valley blocking, in turn, may inhibit propagation of base-level signals and promote formation of relict topography and fluvial hanging valleys.

## Introduction

River incision into bedrock drives landscape change in unglaciated settings and is the key process linking tectonics and topography (Whipple, 2004). However, the process of river incision is sensitive to the amount and caliber of sediment supplied from hillslopes (Sklar and Dietrich, 2001, 2004) such that coarse sediment input can accelerate (Cook et al., 2013)

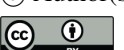



or arrest incision (Scherler et al., 2016). This non-linearity reflects the dual role of coarse sediment in providing abrasive tools and inhibiting incision when deposited on the bed (Gilbert, 1877).

The dependence of river incision on hillslope sediment input also means that river incision, which provides the lower
boundary condition for hillslopes, and landsliding, which delivers coarse sediment to channels, are coupled (Golly et al., 2017; Ouimet et al., 2007). The apparent strength of this coupling, however, varies widely. On the one hand, river aggradation and valley blockages triggered by rock falls and rock avalanches can persist for timescales as long as $10^4$ years (Korup et al., 2006). Similarly, elevated sediment loads following co-seismic landslides can attenuate over timescales as long as centuries (Stolle et al.). These examples suggest that landsliding provides a strong negative feedback on river
incision by causing long lived burial events and hence hiatuses in downcutting (Bennett et al., 2016a; Ouimet et al., 2007). These examples also suggest that valley aggradation or incision following a landslide may occur long after evidence for the landslide itself is recognizable in the landscape. On the other hand, in some settings little evidence of valley blockage is seen despite extremely high rates of landsliding (Korup et al., 2010). In addition, elevated sediment loads following co-seismic landslides can relax in as little as a few years (Hovius et al., 2011). These examples suggest a contrasting view in
which rivers are not strongly perturbed following landslides and in which landsliding occurs essentially passively in response to river incision (Burbank et al., 1996; Larsen and Montgomery, 2012).

There are at least two important factors that govern the sensitivity of rivers to landslide inputs. First, the type and size of landslide dictates the rate and amount of material delivered to channels. This is likely why landslide dam formation is more
common following debris and rock avalanches, which can deliver large volumes of sediment rapidly to rivers (Costa and Schuster, 1988). In addition, valley width is a clear control on landslide dam formation because, all else being equal, a narrow valley can be more easily blocked (Costa and Schuster, 1988). Second, the capacity of a river to transport coarse debris buffers rivers against the effects of landslide inputs. Notably, the study of Larsen and Montgomery (2012) documented landslide rates along the Yarlung Tsangpo River, the most powerful river in the Himalaya (Finlayson et al.,
2002), and found that this location lacks evidence for landslide dams (although evidence for valley blocking moraines is plentiful here) despite extremely high erosion rates (Korup et al., 2010).

While considerable work has been devoted to the problem of landslide dam formation from the perspective of landslide processes (Costa and Schuster, 1988; Korup, 2002), comparatively less work has focused on fluvial controls on the resilience
of rivers to landslide inputs. Accordingly, here we explore what governs river resilience to valley blocking by landslide debris. Towards this end, we exploit two similar geomorphic settings in the California Coast Range where slow-moving landslides, often referred to as earthflows, impinge on river channels with drainage areas that differ by a factor of 30, thereby permitting a comparison of how flow depth, width, and velocity (all of which scale with drainage area) govern river resilience to valley blocking in otherwise similar geomorphic and geologic settings. For two reference sites in each location,



we use high resolution imagery to quantify the size distribution of landslide-derived boulders and then exploit USGS stream gages near each site to quantify the mobile fraction of landslide delivered material for a characteristic 2-year recurrence interval flood. By comparing mapped landslide locations to long profile and valley morphology, we establish locations where valley blocking from earthflows has occurred. Finally, we link the differences in landslide blocking at the two
locations to our analysis of boulder mobility to explore the fluvial controls on resilience to valley blocking.

In this paper we consciously use the term valley blocking instead of damming. While valley blocking from earthflows, like damming, can cause aggradation for kilometers upstream, to depths of several 10's of meters, it is unusual that earthflows deposit sediment rapidly enough to cause the formation of lakes. Hence, to avoid confusion, here we adopt the term valley
blocking.

## 1 Geologic and Geomorphic Setting

The Franciscan Complex is an assemblage of variably deformed and metamorphosed rock units formed in a subduction zone during the Mesozoic and early Cenozoic eras (Wakabayashi, 1992). With widespread occurrence throughout the state of
California (Figure 1), Franciscan lithologies include primarily detrital sedimentary rocks such as sandstones and argillaceous mélanges that are well known for their low strength and high susceptibility to slope failure. Many documented instances of earthflows, in particular, in California occur within these units (Iverson and Major, 1987; Keefer and Johnson, 1983; Kelsey, 1978; Roering et al., 2015; Scheingross et al., 2013). Earthflows are characterized by a flow-like appearance and persistence over decades to centuries (Hungr et al., 2014). They form above fine-grained bedrock in plastic, clayey soil and are
commonly large (>500 m long), deep (>5 m), and move at rates less than ~10 m/a (Baum et al., 2003). Active earthflows frequently extend from ridge-tops to valley-bottoms (Mackey and Roering, 2011) and are classically described as having an 'hourglass' planform outline, with a bowl-shaped source area, an elongate transport zone, and a lobate toe (Keefer and Johnson, 1983). Notwithstanding their name and appearance, most earthflow movement occurs by sliding along discrete basal and lateral shear surfaces (Fleming and Johnson, 1989; Keefer and Johnson, 1983; Simoni et al., 2013; Vulliet and
Hutter, 1988; Zhang et al., 1991). In this paper, we exploit two locations in the California Coast Ranges where earthflows impinge on channels of greatly differing scale. Both of these locations are underlain by the Franciscan Complex lithologic units. Below we describe each location separately.

### 1.1 Arroyo Hondo and Alameda Creek
Arroyo Hondo (200 km$^2$) and upper Alameda Creek (185 km$^2$) drain a rugged region of the northern Diablo Range, northeast of San Jose, California (Figure 1, 2). Their confluence occurs just downstream of Calaveras Dam, which impounds Calaveras Reservoir, the largest reservoir in the San Francisco Bay Area. Where each creek crosses the actively uplifting Diablo Range, it has incised a deep (~ 600 m) canyon into Franciscan formation sandstone and mudstone mélange. The walls of these canyons are draped with earthflows (Figures 2, 3). One of them, Oak Ridge earthflow, was studied by





Nereson and Finnegan (2018), who analyzed its historical motion from air photos that span 1937-2017. Although sliding velocity for the earthflow varied both temporally and spatially, within the persistently active ~ 100 m wide transport zone of the earthflow the mean velocity was 2.15 m/yr Nereson and Finnegan (2018). We use field observations and detailed measurements of boulder size distributions at Oak Ridge earthflow as a reference site from which we make more generalized

inferences about the relationship between earthflows and valley blocking in Alameda Creek and Arroyo Hondo. Here we analyze approximately 20 km of Alameda Creek and 10 km of Arroyo Hondo. These are reaches where the authors have performed extensive field reconnaissance. A USGS gage on Arroyo Hondo is located within the study section considered here (https://waterdata.usgs.gov/usa/nwis/uv?11173200) (Figure 2). At this location, Arroyo Hondo has a drainage area of 200 km$^2$. Annual rainfall at Oak Ridge earthflow is 53 cm, most of which occurs between October and May (Nereson and

Finnegan, 2018).

### 1.2 Eel River

We also exploit a study site developed by Mackey and Roering (2011) along a ~ 30 km section of the main stem Eel River between Dos Rios and Alderpoint (Figure 1, 4) in the northern California Coast Range in Mendocino and Humboldt

counties. At this location, the Eel River cuts a ~ 800 m deep canyon into actively uplifting rocks of the Central Belt of the Franciscan Complex (McLaughlin et al., 2000). The Central Belt consists of mudstone mélange, similar to Arroyo Hondo and Alameda Creek, that surrounds coherent blocks of various lithologies that can be as large as entire mountains (Roering et al., 2015). At this location, Mackey and Roering (2011) tracked the historical motion of 122 earthflows from 1944 to 2006. Over this period, the median annual sliding velocity of all landslides was 0.4 m/yr. More recent work (Bennett et al., 2016b)

has revealed a significant deceleration of these earthflows during the historic California drought from 2012-2015. We analyze river data from a USGS gage located at Fort Seward (https://waterdata.usgs.gov/ca/nwis/uv?site_no=11475000), approximately 12 km downstream of Alderpoint. At this location, the Eel River has a drainage area of 5547 km$^2$. Because there are no major tributary junctions between the study reach and the Fort Seward Gage, we assume the Fort Seward gage is approximately representative of the conditions within the study section. Annual rainfall at Alderpoint is 130 cm (Mackey

and Roering, 2011). We use a reference site at an active earthflow, referred to as the "Mile 201" slide by Mackey and Roering (2001), just downstream from the confluence of Kekawaka Creek with the Eel River (Figure 4). At this location, we make detailed measurements of boulder size distributions being supplied to the Eel River by earthflows via high resolution aerial imagery.

## 2 Methods

### 2.1 Landslide Impacts on Channels

We use measurements of local channel slope (along with floodplain width, described below) to assess the fluvial response to earthflow inputs. Earthflow deposits are commonly comprised of large boulders, leading to steep boulder cascades downstream of landslide blockages, which in turn drive fluvial aggradation upstream (Kelsey, 1978). To highlight such



reaches, we calculate river bed slope over a length-scale that is comparable to the backwater length-scale (~ flow depth/ bed slope) (Paola and Mohrig, 1996; Pfeiffer and Finnegan, 2017). For the Eel River site (as described in more detail later) we use the flow depth calculated for a 2-year recurrence interval flood, 7.1 m, combined with the mean bed slope over the 30 km reach, 0.0026, to calculate a backwater length of ~2700 m. For the Arroyo Hondo site, we use the flow depth calculated

for a 2-year recurrence interval flood, 1.6 m, combined with the mean bed slope over the 30 km reach, 0.016, to calculate a backwater length of ~100 m. Because Alameda Creek has a similar drainage area to Arroyo Hondo at our study location, we calculate slope over a similar length-scale for Alameda Creek. By using the backwater length to calculate slope, our measurements of slope are relatively insensitive to perturbations in river elevation that are smaller than the flow depth, such as those arising, for example, from pool riffle sequences.

Valley aggradation in a steep-walled canyon leads naturally to floodplain widening simply by virtue of the triangular cross-section of a valley (Mey et al., 2015; Reneau and Dietrich, 1991). For this reason, we also make measurements of local floodplain width to complement our channel slope measurements. The logic here is that deep aggradation upstream of landslide blockages should be reflected in local floodplain width. We use LiDAR-derived slope maps to help identify the

slope break that marks the intersection of the steep canyon wall with the low gradient valley bottom alluvial deposits. We hand digitized a line corresponding to this slope break along each side of the three canyon reaches examined here. We then rasterized this line and used ArcGIS to calculate the euclidean distance from both the right and left edges of the floodplain. The sum of these euclidean distance maps within the active floodplain yields an approximation of the local floodplain width, which we extract at each point where we measure elevation.

To highlight potentially landslide impacted river reaches we look for points that mark rapid changes in valley width (from wide to narrow) and rapid changes in river slope (from low to high) (e.g., Ouimet et al., 2007).

For both sites, we take advantage of LiDAR-derived topography data to make measurements of river channel morphology.

For the Eel site, we use LiDAR data as described by Mackey and Roering (2011). For the Alameda Creek/Arroyo Hondo site, we use 1/9 second USGS NED data. We hand digitized thalweg profiles for both study locations using shaded relief maps. We extracted elevation points every 100 m for the Eel River and every 10 m for the Arroyo Hondo and Alameda Creek sites. This difference in spacing reflects the approximate difference in channel width for the two locations.

**2.2 Quantification of Boulder Size Distributions**
We use Google Earth imagery, which we exported to ArcGIS after re-georeferencing, to map boulder size distributions entering channels at the toes of active earthflows at our two reference sites. The toe of Oak Ridge Earthflow is currently collapsing along a series of rotational failures into Arroyo Hondo (Nereson and Finnegan, 2018), which in combination with pre-historic motion of the earthflow toe, has resulted in an accumulation of large, unsorted earthflow-derived boulders in the

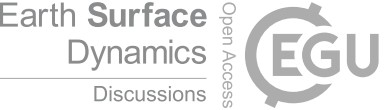

channel of Arroyo Hondo at the base of the earthflow. We digitized all visible boulders (n = 329) as ellipses, which we then

fit with rectangles to quantify the major and minor axis lengths of the boulders. The imagery enable us to identify boulders

down to 30 cm in diameter. For the hydraulic calculations described below, we use the minor axis to approximate the cross-

sectional area of the boulder. Because of the proximity of boulders to the earthflow toe and the absence of sorting in the

field, we treat the distribution of boulders at the toe of Oak Ridge earthflow as representative of the coarse fraction of

material (larger than the 30 cm detection limit) that is eroding out of the earthflow once its fine matrix has been winnowed

away (e.g., Kelsey, 1978). This interpretation is supported by the fact that it is impossible to differentiate individual grains

on bar surfaces upstream of the earthflow toe in aerial imagery. In other words, the bulk of the distribution of bedload that is

moved by the river appears to fall below the detection limit in the aerial imagery.

For the Eel River reference site, we divided the channel into three domains where we digitized boulders separately. Along

the north (river right) bank of the Eel at the toe of the Mile 201 slide is an accumulation of unsorted boulders similar to what

is observed at Arroyo Hondo along the toe of Oak Ridge. However, in contrast to the Arroyo Hondo reference site, within

the thalweg of the Eel River, large boulders are absent and weak sorting is apparent. Hence, we treat the population of

boulders along the north bank (n = 413) as representative of the coarse fraction (> 30 cm) that is eroding out of the mélange

that comprises the body of the Mile 201 earthflow, whereas we consider the thalweg data (n = 602) to be more influenced by

fluvial transport. The south (river left) bank of the Eel River at this site is similar to the north bank. However, we treat the

distribution of boulders from this site (n = 706) separately because we are unsure whether this material is sourced from the

Mile 201 slide or the active earthflow that enters this same location from the south (Mackey and Roering, 2011). Like in

Arroyo Hondo, it is impossible to differentiate individual grains on bar surfaces away from earthflow toes in aerial imagery.

Hence, again, we assume that most of the distribution of bedload that is moved by the river appears to fall below the

detection limit in the imagery.

### 2.3 Quantification of Boulder Mobility

A goal of our analysis is to quantify the mobile fraction of boulders that are delivered from earthflows to channel networks.

However, boulders transported by earthflows in the Franciscan Mélange are commonly comparable or greater in diameter

than the depths of channels into which the earthflows empty. Consequently, below we develop an analytical model to

calculate both the drag force and grain weight of partially submerged boulders in order to create an index of boulder mobility

that is analogous to Shields stress but is physically valid for the case of partial submergence.

The drag force, $F_d$, acting on a boulder for a given mean flow velocity, u, is given by

$$F_d = \frac{1}{2} C_d \rho_f u^2 a \qquad \text{eq. 1}$$

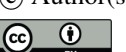



where $C_d$ is a drag coefficient for boulders, estimated to be 0.36 (Bathurst, 1996), $\rho_f$ is the density of water, and a is the cross-sectional area of the submerged portion of the boulder (Appendix 1).

The partially submerged grain weight, $F_g$, is

$$F_g = v_a \rho_s g + v_s (\rho_s - \rho_f) g \qquad\qquad \text{eq. 2}$$

where $v_a$ and $v_s$ are the subaerial and submerged boulder volumes, respectively (Appendix 1), $\rho_s$ is the density of boulders,

and g is the acceleration due to gravity. The ratio of equation 1 to equation 2 is the ratio of the flow-induced drag force acting on a boulder to the partially submerged grain weight.

We note that the definition of Shields stress, $\tau^*$, is also the ratio of drag force acting on a particle relative to the submerged grain weight. Assuming a spherical particle geometry, the bed shear stress, $\tau$, acts on the cross-sectional area, $\pi(D/2)^2$, where

D is, again, particle diameter. Hence, the ratio of drag force acting on a particle relative to the submerged grain weight can be written as

$$\frac{F_d}{F_g} = \frac{\tau \pi \left(\frac{D}{2}\right)^2}{\frac{4}{3}\pi \left(\frac{D}{2}\right)^3 g (\rho_s - \rho_f)} \qquad\qquad \text{eq. 3}$$

which simplifies to

$$\frac{F_d}{F_g} = \frac{\frac{3}{2}\tau}{Dg(\rho_s - \rho_f)} \qquad\qquad \text{eq. 4}$$

an expression that is equivalent to $3/2\tau^*$. Hence,

$$\frac{F_d}{F_g} \sim \frac{3}{2}\tau^* \qquad\qquad \text{eq. 5}$$

For the calculations described in our results, we use a representative flow depth computed from the hydrologic analysis described in the next section. In addition, we assume that boulders have a density of 2650 kg/m$^3$ and are aligned in the most

hydrodynamically stable arrangement in the river such that their major axis is parallel to the flow direction. For the purposes



of the drag calculation, we assume a circular cross-section. However, for the partially submerged weight, we cannot because earthflow derived boulders are rarely spherical. Measured boulders have a ratio of major to minor axis of, on average, 1.5 for the Eel River and 1.6 for Arroyo Hondo. To account for this, we multiply the spherically derived grain weight by the average ratio of major to minor axis length for the two boulder datasets. This assumes that boulders are ellipsoids with

circular cross-sections in one dimension (with a radius corresponding to the measured semi-minor axis) and are elliptical in a plane that is parallel to the flow direction.

## 2.4 Hydrology

The USGS gage on Arroyo Hondo (https://waterdata.usgs.gov/usa/nwis/uv?11173200) is located within the study reach,

roughly 2 km downstream of Oak Ridge earthflow. The gage record includes 36 years of annual peak flood measurements and 248 field measurements of discharge, width and cross-sectional area during both high and low flow events.

The USGS gage on the Eel River (https://waterdata.usgs.gov/ca/nwis/uv?site_no=11475000) is located approximately 25 km downstream of the Highway 201 slide and roughly 12 km downsteam of the edge of the LiDAR data considered here. The

gage record includes 62 years of annual peak flood measurements and 364 field measurements of discharge, width and cross-sectional area during both high and low flow events.

We calculated the recurrence period associated with the annual peak flood measurements for each gage according to standard methods (Dunne and Leopold, 1978). We then estimated the magnitude of the 2-year recurrence interval flood from the

record. To accomplish this, we located the recurrence intervals that bracketed two years in the record and fit a line between these two points. Finally, using linear interpolation we determined the approximate magnitude of the 2-year recurrence interval flood, which we use as a representative high flow event in our analysis. In alluvial channels, the 2-year recurrence interval flood often corresponds to "bankfull" flow (Wolman and Miller, 1960). For this reason, the 2-year flood is commonly interpreted as the "formative" flow with respect to bankfull hydraulic geometry. Here, we use the 2-year flood,

however, simply as a representative flood event that would mobilize the bed in most self-formed alluvial channels, but not necessarily in a channel overwhelmed with landslide debris.

Fortuitously, field measurements of discharge and hydraulic geometry for both gages bracket the 2-year recurrence interval flood. We divide measured discharge by measured cross-section area for each gage record in order to quantify mean flow

velocity for each measurement. We also divide measured cross-section area by width to quantify mean flow depth by assuming a rectangular geometry. We then plotted mean flow depth versus discharge and mean flow velocity versus discharge for each record. The relationships between discharge and velocity and discharge and flow depth were both well fit with power-law relationships for the Eel River record. Hence for this record, we simply use the best-fitting power law relationship to find the mean velocity and mean depth associated with the two-year recurrence interval flood. For Arroyo



Hondo, a power law does not fit the relationships between discharge and velocity and discharge and flow depth at the discharges near the two year recurrence interval flood. Consequently, for this record we used a linear fit for discharges greater than 20 m³/s, which fits the data well in this region. We then apply this linear fit in order to determine mean flow depth and mean velocity associated with the 2-year recurrence interval flood on Arroyo Hondo.

### 2.5 Landslide Identification

For the Eel River site, we use the landslide mapping of Mackey and Roering (2011), who identified 122 active individual earthflows within the Eel River study site using historical aerial photos (Figure 4). More recent studies using optical images and radar interferometry show that the majority of these landslides are still active (Bennett et al., 2016a; Bennett et al., 2016b; Handwerger et al., 2015).

For Arroyo Hondo and Alameda Creek we used a combination of airborne synthetic aperture radar interferometry (InSAR), LiDAR topography, and field reconnaissance to identify slow landslides that are either currently active, or have been active in the recent geomorphic past.

Based on our own field reconnaissance in the area, as well as through interpretation of LiDAR-derived topographic maps, we have identified several large earthflows within the field area that clearly impinge on the channels of Arroyo Hondo and Alameda Creek (Figure 2, 3). We also process radar interferometry data from the airborne NASA Uninhabited Aerial Vehicle Synthetic Aperture Radar (UAVSAR) platform to identify active landslides between 2009 and 2017 (Appendix 2).

20 UAVSAR operates with a L-Band radar wavelength (~ 24 cm) and collects data at this location approximately 2 times per year along track 23503 (aircraft moving at heading 230 deg and looking at 140 deg). We processed 31 interferograms using the InSAR Scientific Computing Environment (ISCE) software package developed at JPL/Caltech and Stanford (Rosen et al., 2012). We remove topographic contributions to the phase using a 12 m DEM from the DLR TanDEM-X satellites. We also reduce InSAR phase noise using a standard power spectral filter with a value of 0.5 (Goldstein and Werner, 1998).

25 Finally, we selected 7 high quality interferograms (minimal unwrapping errors, high coherence) to compute average line-of-sight (LOS) velocity map for landslides within the study area.

### 3 Results

#### 3.1 Long Profile and Valley Width

30 Figures 5A, 6A, and 7A show elevation long profiles of Arroyo Hondo, Alameda Creek, and the Eel River, respectively. Also indicated on the figures are the locations where landslides intersect river channels, as shown in Figures 2-4. For the Eel River, landslide locations come from Mackey and Roering (2011). For Alameda Creek, landslide locations are, again, based on a combination of field reconnaissance, LiDAR interpretation, and InSAR.





Figures 5B-C and 6B-C show measured channel slope and valley width (normalized to the mean for each river) for Alameda Creek and Arroyo Hondo. For both channels, mapped landslide locations coincide with locations along the river marked by rapid changes in valley width from wide upstream to narrow downstream and from low slope upstream to high slope downstream. We also note that the sharpest river slope increase associated with earthflows along Alameda Creek and

Arroyo Hondo occurs at Oak Ridge earthflow, which is the only failure that is both currently active and coupled to a river channel (Arroyo Hondo) according to InSAR (Figure 2,3) and feature tracking (Nereson and Finnegan, 2018).

In contrast, Figures 7B-C show that active landslides along the Eel River have no obvious impacts on long profile slopes and valley widths. At no location does the channel slope increase by a factor of two in the study region, and except for one spot

along the Eel River section, valley bottom width is also always less than a factor of two greater than the mean.

**3.2 Boulder Size Distributions**

Figure 8 shows empirical cumulative distribution functions (CDFs) of the intermediate axis of boulders mapped at the toe of Oak Ridge Earthflow, and for the right and left banks of the Eel River at the toe of the Mile 201 slide. Although the

distributions diverge below ~ 1 m, the curves are quite similar for boulders larger than 1 m. A two-sample Kolmogorov-Smirnov test for the portions of each of the three measured boulder distributions above 1 m is unable to reject the hypothesis (at the 5% significance level) that the three boulder populations are drawn from the same distribution.

**3.3 Hydrology**

Figures 9A and 10A show recurrence interval plotted against discharge for measured peak flows for the two reference gages on Arroyo Hondo and the Eel River. Also shown in Figures 9B-C and 10B-C are field measurements of mean depth and velocity, respectively, for the two gages plotted as a function of discharge. As noted earlier, the two year recurrence interval event falls within the range of field measurements for the two sites, making accurate interpolation of the characteristic two year recurrence interval depths and velocities straightforward. Table 1 reports these characteristic values

**3.4 Quantification of Boulder Mobility**

Using equations 1-2, we calculate the mobility of the range of boulder sizes for the two reference gage sites. The results of this analysis are plotted in Figure 11A, which shows that for a two year recurrence interval flood, the boulder mobility threshold for the Eel River is ~ 4.9 m, which is approximately the 99th percentile grain size for landslide derived boulders.

For Arroyo Hondo, the boulder mobility threshold is ~ 2.4 m, which is approximately the 90th percentile grain size for landslide derived boulders. These calculated mobility thresholds are also plotted on Figure 8. Given that the choice of a specific mobility threshold is somewhat arbitrary, we also plot in Figure 11B the relative mobility of boulders in the Eel River compared to Arroyo Hondo. This analysis shows that depending on grain size, boulders in the Eel are ~1.8-2.6 times more easily mobilized than in Arroyo Hondo for a two-year recurrence event.





Notably, although only the upper 10% of the CDF for Arroyo Hondo is immobile, this portion of the distribution nevertheless represents approximately 80% of the total volume of landslide material shed into Arroyo Hondo (Figure 12). On the Eel River, the immobile upper 1% of the CDF of boulder sizes, by contrast, represents only approximately 20% of the

volume of landslide material shed into the Eel River (Figure 12). This dramatic contrast in the volume of immobile material that is shed into each of the two channels is illustrated in Figures 13 and 14, which identify in satellite imagery the individual boulders that are above the 2-year recurrence interval mobility threshold for the two reaches, calculated from equations 1-2. Whereas on the Eel River (Figure 14), immobile boulders are sparsely distributed and generally confined to the edges of the channel, on Arroyo Hondo the immobile boulders comprise clusters that span the entire channel (Figure 13).

## 4 Discussion & Conclusions

### 4.1 What Controls Valley Blocking?

Analysis of valley widths and river long profiles in Alameda Creek and Arroyo Hondo shows a very consistent picture in which landslides that intersect the channel force tens of meters of gravel aggradation for kilometers upstream, leading to

apparently long-lived sediment storage and channel burial at these sites (Figures 5A-C, 6A-C). In contrast to Arroyo Hondo and Alameda Creek, the Eel River does not display knickpoints at or aggradation upstream of locations where earthflows impinge on the channel, such as at the Highway 201 slide (Figure 7A-C).

Analysis of satellite imagery shows that immobile earthflow-derived boulders are sparsely distributed and generally confined

to the edges of the channel on the Eel River (Figure 14), whereas immobile earthflow-derived boulders comprise clusters that jam the entire channel on Arroyo Hondo (Figure 13). This striking difference in channel morphology implies that the Eel River is apparently capable of transporting most earthflow derived blocks, whereas Arroyo Hondo is not. This inference is supported by our mobility analysis, which shows that 80% of the volume of coarse landslide material delivered to Arroyo Hondo is immobile, compared to only 20% on the Eel River (Figure 12).

The contrasting long profile and channel morphology of the two reference locations suggests that the formation of valley blockages by earthflows is very sensitive to the mobility of the coarsest fraction of sediment. In this way, landslide impacted channels may act like step-pool channels, where channel spanning boulder jams locally impede coarse sediment transport. According to flume experiments (Zimmermann et al., 2010) as well as experiments of granular flow through hoppers (To et

al., 2001) and silos (Zuriguel et al., 2005), particle jams become much more likely when the channel or aperture width is approximately five grain diameters or less. Not coincidentally, bulk friction angle (and hence stability) of chains of clasts is larger for smaller chains, with an inflection in the relationship between frictional stability and number of clasts at around 5 grains (Booth et al., 2014). Notably, the ratio of channel width to the threshold diameter for immobile boulders on Arroyo Hondo is ~5 as opposed to ~25 for the Eel River, suggesting that Arroyo Hondo and Alameda Creek are in a regime where



they are much more susceptible to jamming. Moreover, once organized into jams, boulders are much harder to dislodge than when they are isolated on the bed (Prancevic and Lamb, 2015), suggesting that the process of valley blocking by boulders is governed by a positive feedback process.

Taken together, the results of our analysis suggest that valley blocking by earthflows is a process that is controlled by the clustering of a few boulders in a narrow channel, which can nevertheless impede coarse sediment transport, bedrock incision and trigger aggradation for km's upstream. Although we have no direct constraints on the process of landslide damming in other settings, particularly where catastrophic landslides are more common, it seems reasonable that the same processes that we identify here may also apply. Indeed, Costa and Schuster (1988) showed that valley width is a clear control on landslide

dam formation because it is easier to block a narrow valley than a wide valley. In addition to this geometric effect, we speculate that channel spanning boulder jams, which will be more prevalent in narrower valleys, may also contribute to the tendency for landslide dams to form in narrow valleys.

### 4.2 Implications for River Incision and Landscape Evolution

As noted in the introduction, the dependence of river incision on hillslope sediment input means that river incision, which provides the lower boundary condition for hillslopes, and landsliding, which delivers coarse sediment to channels, are coupled (Golly et al., 2017; Ouimet et al., 2007). The apparent strength of this coupling, however, varies widely. The results of our analysis imply that the Eel River, and larger rivers like it, should be able to sustain vertical incision despite active earthflows. In other words, in settings where landslide-derived blocks are mobilized by large floods, landsliding may not

represent a strong negative feedback on river incision. In contrast, in settings such as Arroyo Hondo and Alameda Creek, where most of the landslide-derived material is too large to be mobilized, landsliding should represent a strong negative feedback on vertical river incision by triggering channel spanning boulder jams that force aggradation over large sections of river upstream of active landslides (Figures 5A-C, 6A-C).

River incision at the toe of hillslopes, which both steepens hillslopes and debutresses them, is a trigger for earthflow motion (Bennett et al., 2016a; Bilderback et al., 2015; Golly et al., 2017; McKean, 1993; Nereson and Finnegan, 2018). Thus, aggradation upstream of valley blockages may indirectly hinder active landsliding by suppressing vertical incision. In addition, deep aggradation may directly cease landsliding by burying and hence buttressing the toes of active landslides by adding an additional longitudinal force that resists sliding. Indeed, buttressing of slow landslides by valley bottom

aggradation is a potentially important but underappreciated process that can contribute to landslide stability in some settings (Johnson et al., 2016).

In addition to base-level forcing by rivers, both climate (Bennett et al., 2016b; Mackey et al., 2009; Nereson and Finnegan, 2018) and sediment supply (Mackey and Roering, 2011; Nereson and Finnegan, 2018) also govern the motion of slow



landslides by modulating stresses via pore fluid pressures and landslide thickness, respectively (Iverson, 1986). For this reason it can be difficult to directly attribute landslide activity (or its absence) to a specific forcing. That said, a notable difference between our two sites is that the Eel River site is characterized by dozens of active earthflows along the river canyon (Figure 4), whereas over a comparable length of river in Alameda Creek and Arroyo Hondo only two of the many

mapped landslides are currently or historically active (Figures 2, 3), and only one is directly coupled to a river channel. It's possible that this difference reflects either the relatively drier climate of the San Francisco Bay area compared to Mendocino County or regional differences in rates of river incision and hence base-level forcing. However, it's also possible that the relative inactivity of slow landslides in Alameda Creek and Arroyo Hondo reflects the fact that valley blocking by earthflows has triggered aggradation and hence stabilized landslides over the length of much of these rivers. An obvious area of future

research would be to constrain the lifespan of valley blockages by earthflows as this will more directly inform the strength of the negative feedback discussed above.

Finally, our results imply that the lower drainage area, upper portions of earthflow-dominated catchments may be particularly prone to blocking because they lack the flow depths sufficient to transport landslide debris. By inhibiting the

upstream propagation of base-level signals, valley blocking earthflows may therefore promote the formation of so-called "relict topography" (Clark et al., 2006; Schoenbohm et al., 2006) where the upper portions of watersheds are unable to incise at the same rate as the mainstream. Indeed, Bennett et al. (2016a) noted that tributaries to the Eel River are choked with coarse sediment that significantly impedes river incision into bedrock, and hypothesized that earthflow dominated catchments are prone to a so-called "landslide cover effect," which prevents or delays the upstream propagation of base-level

signals, thereby leading to the formation of relict topography. Similarly, Korup et al. (2010) argued that sediment inputs from rockfalls and glaciers have suppressed river incision into the margin of the Tibetan Plateau, aiding in its apparent longevity.

Our results are generally supportive of these perspectives and offer a simple mechanism for the instability that triggers the

incisional shutdown of earthflow dominated channels. Moreover, because we identify a clear discharge (and thus drainage area) dependence on the susceptibility of channels to jamming from earthflow derived boulders, our results imply that tributary junctions are likely to mark boundaries between relict topography and actively incising canyons. In other words, landslide or debris flow derived boulder jams in narrow channels provide an alternative explanation for the phenomenon of fluvial hanging valleys, where tributary channels in steep canyons are apparently unable to incise at the rate of trunk channel

incision (Crosby et al., 2007; Wobus et al., 2006).

We note that Shobe et al. (2016) suggested that detachment-limited channels where boulders are fed from hillslopes will naturally evolve to have greater stresses in order to compensate the extra drag from boulders. Finnegan et al. (2017) also showed a clear relationship between bedrock channel slope and grain diameter and suggested it was due to the transport-




limited nature of erosion in the setting they examined. Finally, Ouimet et al. (2007) argued that channels that are prone to landsliding should evolve to be steeper because of the decreased erosional efficiency associated with landslide burial (essentially the "landslide cover effect" of Bennett et al., 2016a).

Given these results, it is tempting to interpret the difference in boulder transport capacity on the Eel River relative to Arroyo Hondo and Alameda Creek from the perspective of these studies, which would imply that the Eel River has evolved to compensate earthflow boulder fluxes, whereas Arroyo Hondo and Alameda Creek have not. However, we note that bankfull stresses are actually very similar on Arroyo Hondo compared to the Eel River (Table 1). This serves to illustrate the fact that it is the Eel River's deep flows, not its slope, that enables it to move very large boulders, as is clear from equation 1, where

the cross-sectional area of the boulder exposed to the flow (which is a function of flow depth) governs the boulder drag force. Because the flow depth is largely determined by drainage area (Leopold and Maddock Jr, 1953), it cannot respond in a simple way to sediment supply. Hence, our results suggest that the bedrock channels studied here have not evolved in an obviously identifiable way to boulder inputs. Rather, our results imply that the capacity (or lack thereof) to move earthflow-derived boulders is largely an accident of drainage area. Notably, we have observed epigenetic gorges (Ouimet et al., 2008))

at several locations where active earthflows impinge on Alameda Creek, which suggests that channels in the Franciscan Mélange may simply incise around the margins of episodically-delivered (e.g., Mackey and Roering, 2011) earthflow boulder deposits. Given the extremely weak matrix of the Franciscan Mélange, we speculate that the extra bedrock incision that is necessitated by period epigenetic gorge formation may have little impact on the time-integrated erosional efficiency of these channels. That said, without much more detailed field studies, we can only conjecture.

**Acknowledgements**

This work was supported by a National Science Foundation (NSF) Graduate Research Fellowship awarded to A.L.N. and the Geomorphology and Land Use Dynamics Program of NSF (EAR–1658800 and EAR–1613122 to N.J.F.). A.L.H's research was supported by an appointment to the NASA Postdoctoral Program at the Jet Propulsion Laboratory, administered by

Universities Space Research Association under contract with NASA. We thank the San Francisco Public Utilities Commission and Russ Fields for site access.





**Appendix 1 - Submerged Cross-Sectional Area and Volume of Boulders**

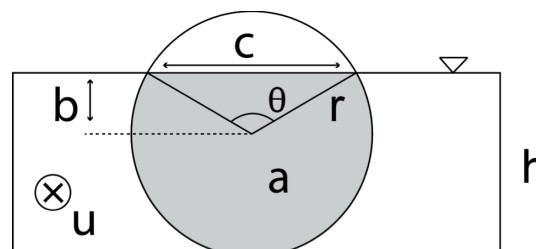

Figure A1 provides a sketch of the geometry employed in the model. In Figure A1, $h$ is flow depth, $u$ is mean flow velocity into the page, $a$ is the submerged cross-sectional area of the boulder (shown by shaded region), $r$ is boulder radius.

b, is given by:

$$b = \frac{D}{2} - (D - h)$$
eq. 1a

20    for the case where h >= D/2, where h is flow depth and D is boulder diameter. Alternatively, for the case where h < D/2, b is given by

$$b = h$$
eq. 1b

25    c is then calculated via the pythagorean theorem:

$$c = 2\left(\left(\frac{D}{2}\right)^2 - b^2\right)^{\frac{1}{2}}$$
eq. 2a

for the case where h >= D/2. Alternatively, for the case where h < D/2, c is given by





$$c = 2\left(\left(\frac{D}{2}\right)^2 - \left(\left(\frac{D}{2}\right) - b\right)^2\right)^{\frac{1}{2}}$$   eq. 2b

$\theta$, measured in radians, is given by,

5   $$\theta = \sin^{-1}\left(\frac{c}{D}\right)$$   eq. 3

Finally, the area of the submerged portion of the boulder, a, is given by

$$a = \pi \left(\frac{D}{2}\right)^2 - \left(\frac{1}{2}\left(\frac{D}{2}\right)^2 (\theta - \sin\theta)\right)$$   eq. 4a

for the case where h >= D/2.  Alternatively, for the case where h < D/2, a is given by

$$a = \frac{1}{2}\left(\frac{D}{2}\right)^2 (\theta - \sin\theta)$$   eq. 4b

15   Finally, for the case where h > D, a is given simply by

$$a = \pi \left(\frac{D}{2}\right)^2$$   eq. 4c

In order to calculate the partially submerged grain weight, $F_g$, we calculate the submerged and sub-aerial boulder volumes
20   separately.

For the case where h >= D/2, the subaerial boulder volume, $v_a$, is given by

$$v_a = \frac{1}{3}\pi(D - h)^2 \left(3\left(\frac{D}{2}\right) - (D - h)\right)$$   eq. 5

The submerged volume, $v_s$, is then given by

$$v_s = \frac{4}{3}\pi \left(\frac{D}{2}\right)^3 - v_a$$   eq. 6





Where $\rho_s$ represents boulder density and g is the acceleration due to gravity. For the case where h < D/2, $v_s$ is given by

$$v_s = \frac{1}{3}\pi(D-h)^2\left(3\left(\frac{D}{2}\right)-(D-h)\right)$$

eq. 7

5    and $v_a$ is given by

$$v_a = \frac{4}{3}\pi\left(\frac{D}{2}\right)^3 - v_s$$

eq. 8

**Appendix 2. UAVSAR Acquisitions and InSAR Pairs**

| UAVSAR Acquisition | 20090220 | 20091120 | 20100511 | 20100115 | 20100116 | 20111114 | 20121102 | 20130507 |
|---|---|---|---|---|---|---|---|---|
| 20090220 | | ▓ | | ▓ | | | | |
| 20091120 | | | ▓ | | | | | |
| 20100511 | | | | | ▓ | | | |
| 20100115 | | | ▓ | | | | | |
| 20100116 | | | | | | | | |
| 20111114 | | | | | | | ▓ | |
| 20121102 | | | | | | | | ▓ |
| 20130507 | | | | | | | | |



| River | Two Year Discharge ($m^3$/yr) | Two Year Mean Flow Depth (m) | Two Year Mean Velocity (m/s) | Slope | Two Year Shear Stress (Pa) | Drainage Area ($km^2$) |
|---|---|---|---|---|---|---|
| Eel River | 2826 | 7.1 | 3.0 | 0.0026 | 180 | 5547 |
| Arroyo Hondo | 92 | 1.6 | 2.4 | 0.016 | 250 | 200 |

**Table 1: Channel characteristics for the two reference locations.**




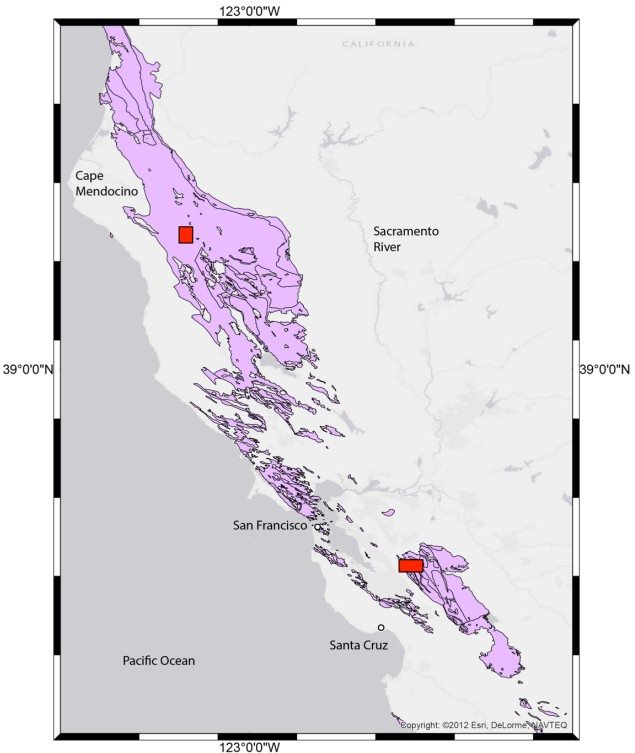

**Figure 1: Overview of the study region. Franciscan Complex rocks are shown in purple. Red boxes indicate the locations of the two field locales.**





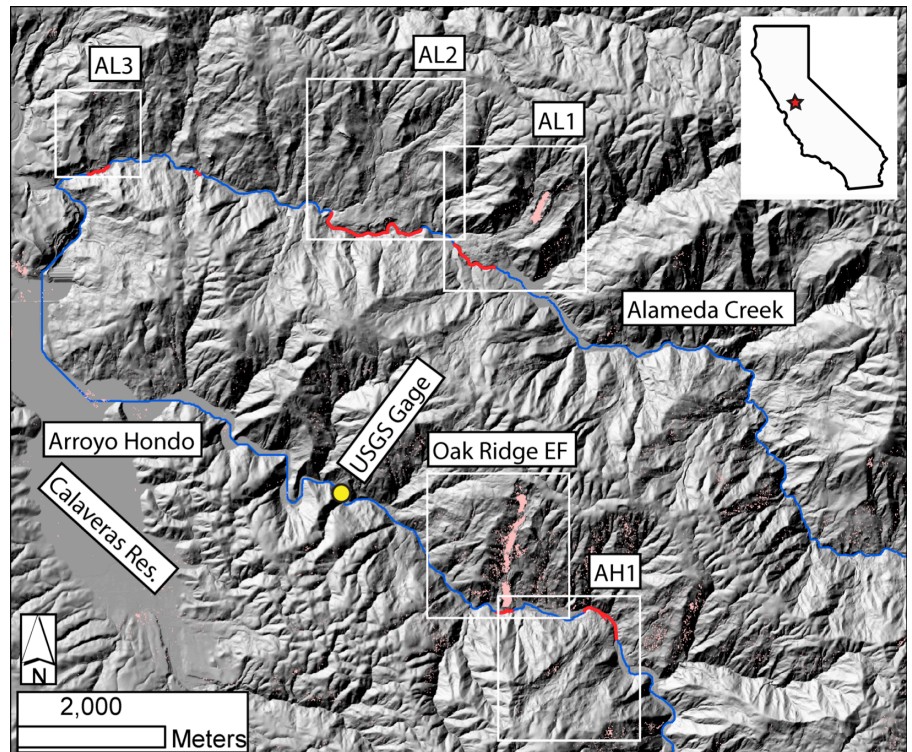

5  **Figure 2: Shaded relief map of the study reaches of the Alameda Creek and Arroyo Hondo watersheds. White boxes highlight large mapped earthflows in the study area, which are shown in more detail in Figure 3. Rivers channels are indicated with blue lines, except where they intersect active earthflows (red lines). Areas of light red shading show regions where the InSAR analysis indicated line-of-site velocities in excess of 3 cm/yr.**



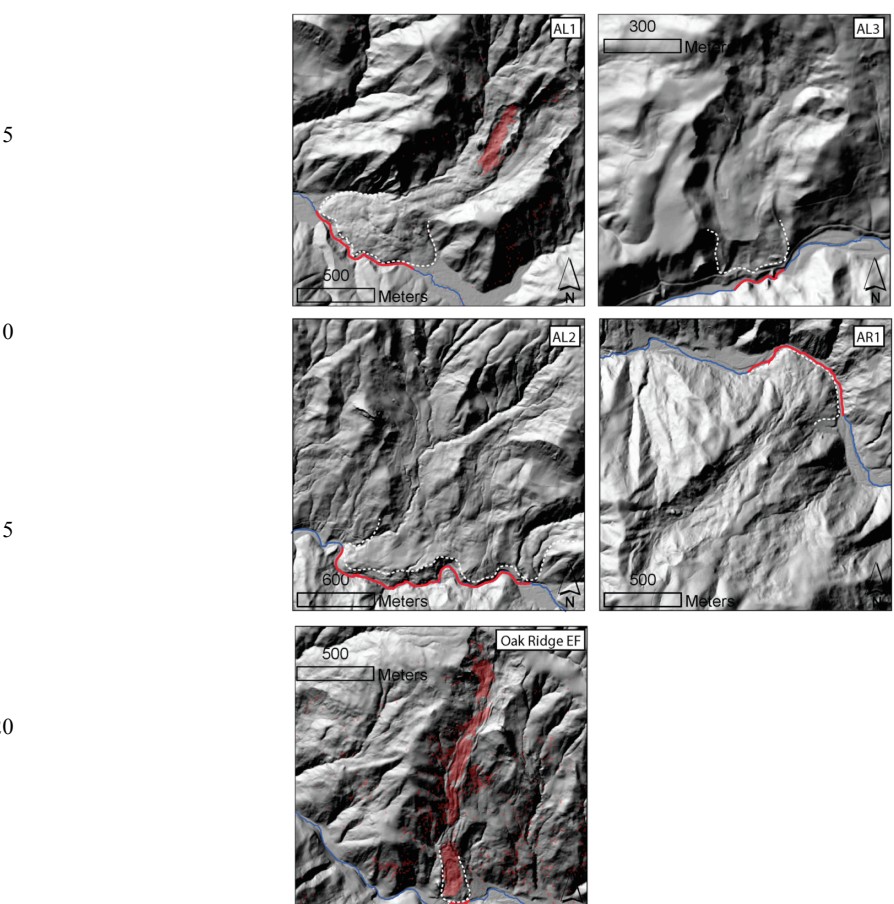

**Figure 3:** **Shaded relief maps of three large earthflows on Alameda Creek (A-C) and two on Arroyo Hondo (D-E).**
**White dashed lines indicated the mapped edge of earthflow toes, and red lines indicate where river channels (shown**
**in blue) intersect earthflows. Areas of red shading show regions where the InSAR analysis indicated line-of-site**
**velocities in excess of 3 cm/yr.**



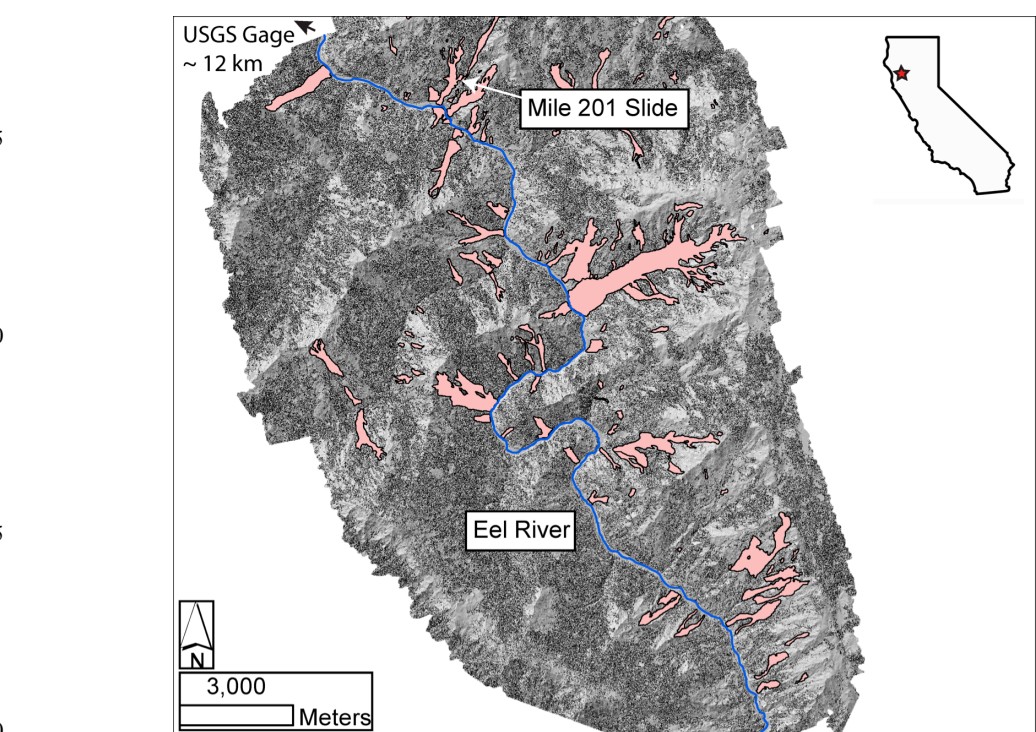

**Figure 4: Shaded relief map of the Eel River site. The thalweg of the Eel River is shown in blue. Areas of red shading are active earthflows mapped by Mackey and Roering (2011). River flow is from the lower right to upper left.**





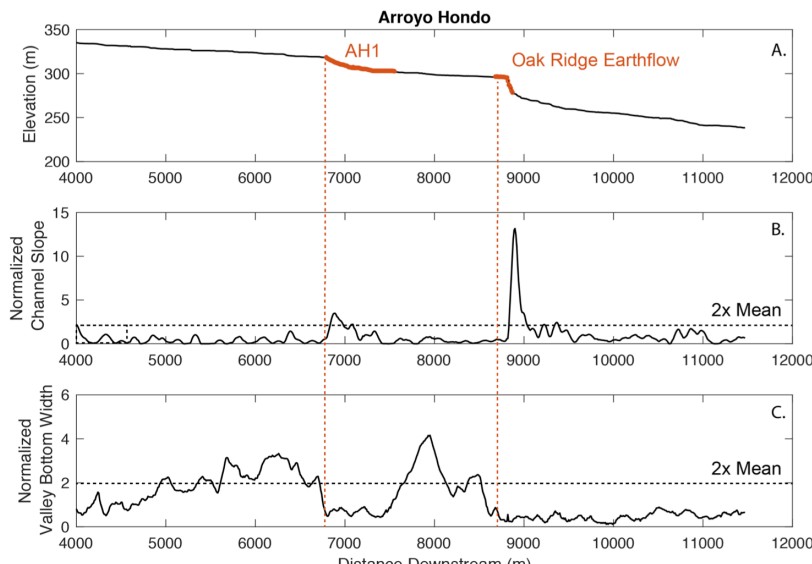

**Figure 5: A) Elevation long profile of Arroyo Hondo. Locations where the channel intersects mapped earthflows in Figures 2 and 3 are shown in red. B) Channel slope on Arroyo Hondo normalized by the mean slope over the reach shown in Figure 5A. C) Valley bottom width Arroyo Hondo normalized by the mean valley bottom width over the reach shown in Figure 5A. In A-C, the vertical dotted red lines highlight the upstream edge of mapped earthflows.**

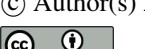



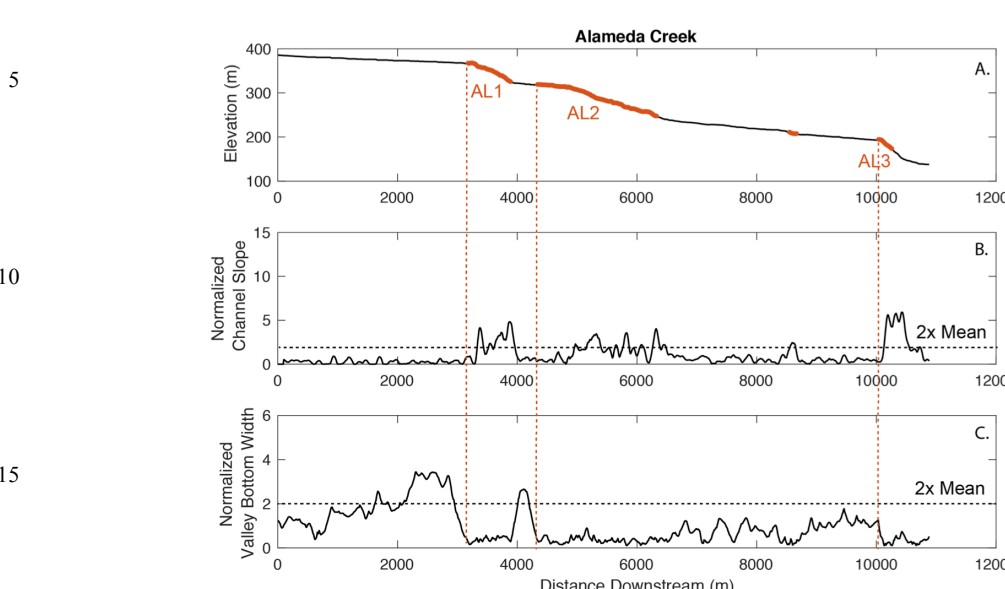

**Figure 6: A) Elevation long profile of Alameda Creek. Locations where the channel intersects mapped earthflows in Figures 2 and 3 are shown in red. B) Channel slope on Alameda Creek normalized by the mean slope over the reach shown in Figure 6A. C) Valley bottom width Alameda Creek normalized by the mean valley bottom width over the reach shown in Figure 6A. In A-C, the vertical dotted red lines highlight the upstream edge of mapped earthflows.**





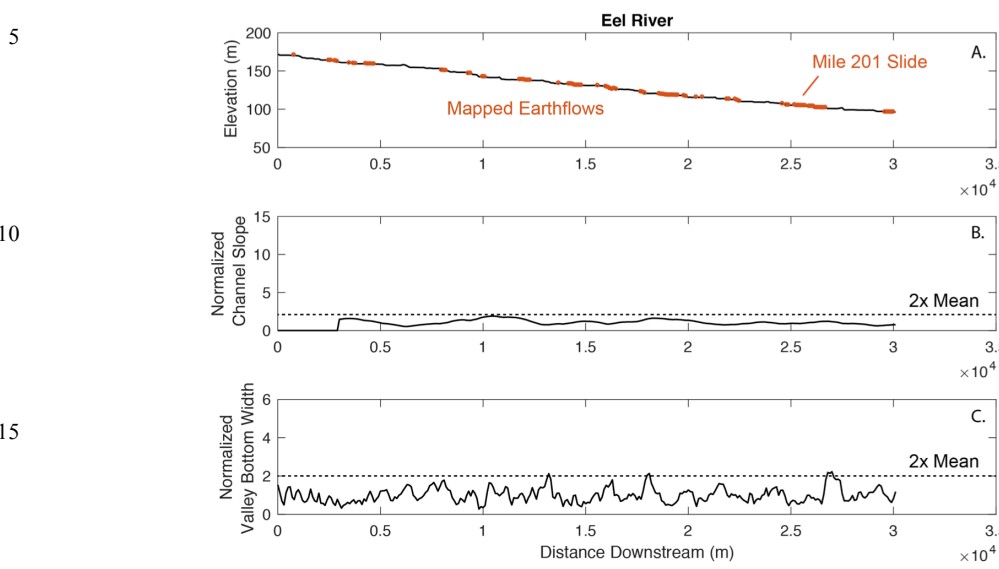

**Figure 7: A) Elevation long profile of the Eel River. Locations where the channel intersects mapped earthflows in Figure 4 are shown in red. B) Channel slope on the Eel River normalized by the mean slope over the reach shown in Figure 7A. C) Valley bottom width on the Eel River normalized by the mean valley bottom width over the reach**
25 **shown in Figure 7A.**





**Figure 8: Empirical cumulative distribution functions for the intermediate axes of earthflow-derived boulders at the toe of Oakridge Earthflow and for the banks of the Eel River site at the Mile 201 Slide. The vertical lines indicate the threshold mobile grain diameter for two year recurrence interval floods at the two reference sites.**





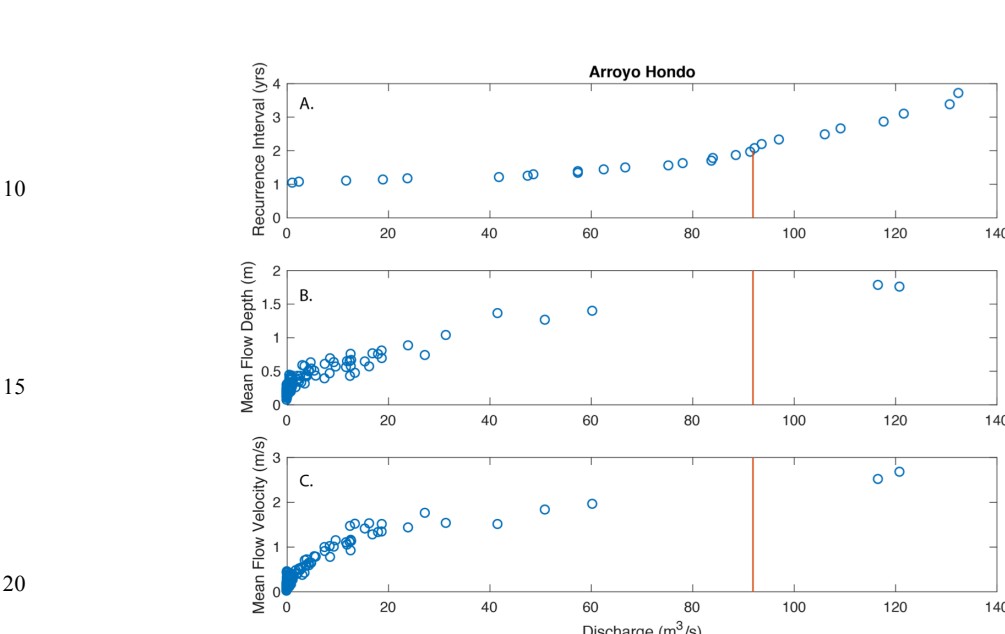

**Figure 9: A) Recurrence interval, B) measured flow depth, and C) measured mean velocity versus discharge for the**
25 **USGS Arroyo Hondo gage. Red vertical lines indicate the two year recurrence interval flood magnitude.**

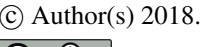



25  **Figure 10:  A) Recurrence interval, B) measured flow depth, and C) measured mean velocity versus discharge for the USGS Eel River gage at Fort Seward.  Red vertical lines indicate the two year recurrence interval flood magnitude.**




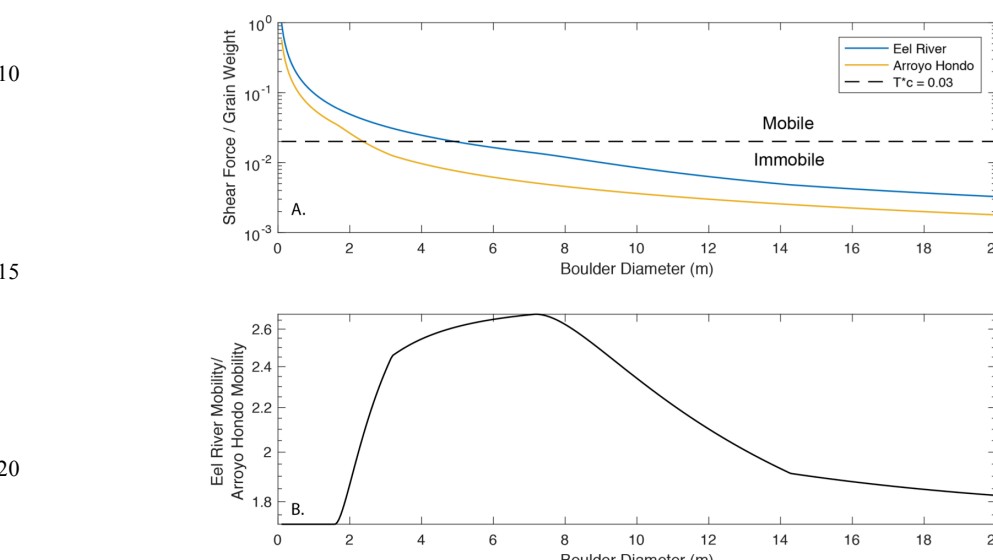

25 **Figure 11: A) Ratio of drag force to grain weight versus boulder diameter for two year recurrence interval floods on the Eel River and Arroyo Hondo. The horizontal dashed line is equivalent to a critical shields stress ($\tau^*_c$) of 0.03, which we use to define the mobility threshold. B) Mobility of boulders on the Eel River relative to Arroyo Hondo for a two-year recurrence interval flood.**







**Figure 12: Cumulative volume fraction as a function of boulder diameter for the Mile 201 and Oakridge Earthflow sites. The dashed lines indicate the cumulative volume of material that is mobile in a 2-year recurrence interval flood for the Eel River and Arroyo Hondo.**



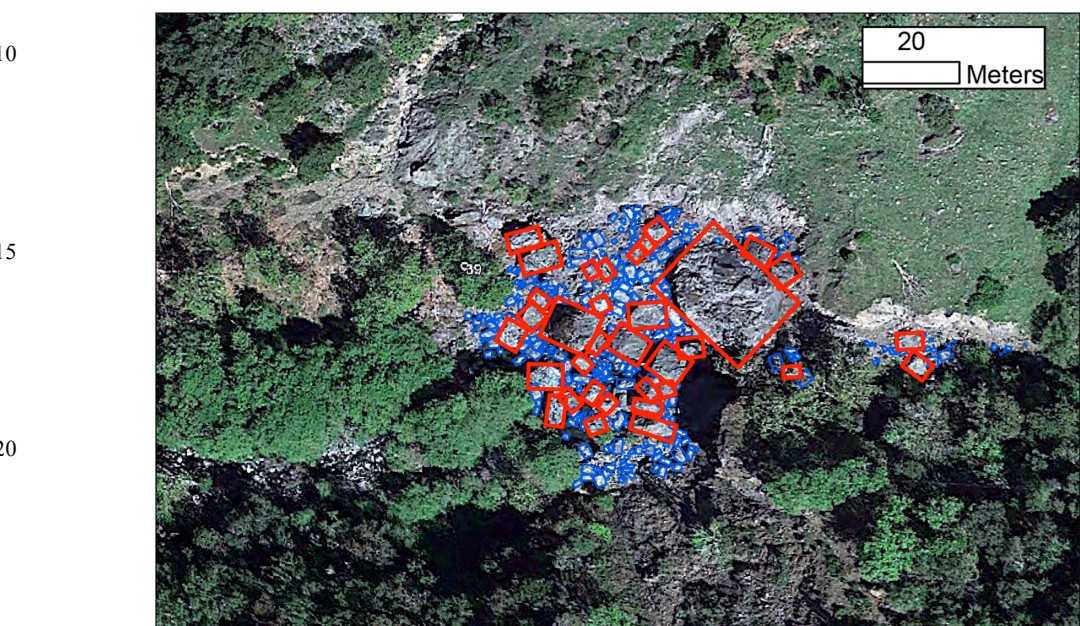

**Figure 13:** **Satellite image of Arroyo Hondo at the toe of the Oakridge earthflow. Red boxes outline boulders that, according to our analysis, are immobile in a 2-year recurrence interval event. Blue boxes outline boulders that, according to our analysis, are mobile in a 2-year recurrence interval event**



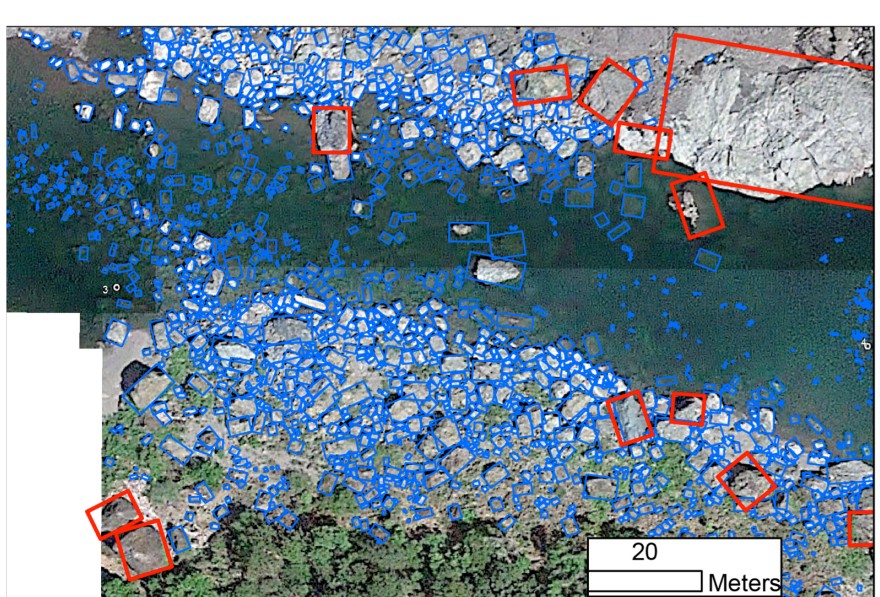

**Figure 14:** **Satellite image of the Eel River at the toe of the Mile 201 earthflow. Red boxes outline boulders that, according to our analysis, are immobile in a 2-year recurrence interval event. Blue boxes outline boulders that, according to our analysis, are mobile in a 2-year recurrence interval event**



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
