# Peer review of "River Channel Width Controls Blocking by Slow-moving Landslides in California's Franciscan Mélange"

_Earth Surface Dynamics, 2018_

## Referee Comment (RC1) · Prancevic (Referee) · 5 Dec 2018

This manuscript addresses the interesting question of how different parts of the river system deal with earthflow-derived boulders and how this could influence hillslope-channel feedbacks and knickpoint propagation. The manuscript is well written and was a pleasure to read. However, I feel that the manuscript has several issues that should be addressed prior to publication.

In the manuscript, the authors present a mechanistic argument for why small rivers are more susceptible to blocking by earthflows than are large rivers. The argument hinges on the tendency for small rivers to have both narrower widths, making them more sus-

ceptible to boulder jamming, and shallower flow depths, leaving large boulders emergent from the flow during bankfull discharges. Both of these mechanisms should lead to boulders being more stable, locally inhibiting bedrock incision with measurable impacts on the river longitudinal profile: local steepening of the river where it crosses the toe of the earthflow, and widening of the river upstream. The manuscript argues that these topographic indicators of boulder stability are absent in the longitudinal profiles of large rivers because boulders are more easily transported away.

These arguments seem very reasonable to me, qualitatively, but I see some issues with the evidence used to verify these mechanisms in this manuscript:
* * *
Flow hydraulics and flow competence: My primary concern with the manuscript is the line of reasoning used to calculate the maximum boulder size that can be transported by the Eel River and Arroyo Hondo. First, as far as I understand, the analysis relies on hydraulic measurements (flow depths, velocities, and discharges) that were made at USGS gage stations that are 2 km (for Oak Ridge earthflow) and 25 km (Mile 201 slide) downstream of the earthflow deposits. Flow depths, velocities, and discharges at the gage sites are not the same as those at the earthflow deposits. In particular, flow depth and velocity are both sensitive to changes in bed sediment size and channel width. The presence of coarse sediment (such as that found in the earthflow deposits) tends to reduce flow velocities and increase flow depths (e.g., Rickenmann and Recking, 2011). This phenomenon is particularly concerning for this analysis because it may reduce the tendency for boulders to emerge from the flow, which the manuscript proposes as the primary mechanism to stabilize boulders. Second, I am not sure that it is necessary to introduce a new, untested calculation for boulder mobility when tested models already exist.

Simultaneously accounting for the effect of boulders on sediment mobility and flow resistance (for calculation of flow velocity and depth) is tricky and prone to error, but these difficulties can be circumvented by using a critical dimensionless stream power
to calculate flow competence (Parker et al., 2011). This criterion has been shown to be a more reliable predictor of sediment mobility than the critical Shields stress in shallow, rough flows (Parker et al., 2011; Ferguson, 2012; Prancevic and Lamb, 2015), and it only requires information about discharge, width, slope, and grain size, all of which the authors have measured at the earthflow deposits.

In a quick calculation, I found that this method predicts that the largest mobile sediment during the bankfull flood is ∼20 cm for both sites. I may have missed something, but this would suggest that there is not a large difference in flow competence between the two sites. Moreover, these grain sizes seem much more reasonable to me than 2.4 m and 4.9 m. (Can a 4 m boulder really be transported by a 2-year flood in the Eel?) This estimate of the mobile size fraction is also consistent with the observation that there are very few boulders larger than 30 cm found in the river outside of earthflow deposits at both sites. ____________________

Comparison of longitudinal profiles: I do think that the longitudinal profiles (Figs. 5-7) show different earthflow signatures between the study sites, but I think that the comparison would be more compelling if it were more even. The profile of the Eel River is 3x to 4x longer than the other two profiles but is squeezed into the same plot size, which makes it difficult to compare the topographic imprint of earthflows (which seem to not change substantially in size between the two study areas). Moreover, the slope measurements on the Eel River were made at a resolution 10x larger than at the other study sites, potentially smoothing over some of the variability in slope.

I think it would be more convincing to use the same spacing for slope measurements and to zoom in on a portion of the Eel River profile such that the profile examined a similar length to the others two. ____________________

Boulder supply rate: The manuscript focuses on the relative size distributions of coarse sediment supplied to the river between the two study areas. However, it may also be more important to consider the supply rate of coarse boulders. If boulders are delivered

very slowly, then the river can rely on bigger, rarer flood events to remove the boulders. It may be outside of the scope of this paper to consider this effect quantitatively, but it should at least be discussed, especially because the Oak Ridge earthflow seems to be sliding 5x faster than the Mile 201 slide. This could be part of the explanation for why there are more big boulders in the river next to the Oak Ridge earthflow.

Also, if the supply rate is important, it is not only the distribution of coarse sediment delivered by the earthflow that matters, but also the proportion that is coarse sediment. What portion of earthflow-derived material is just fines that is just being easily washed away, and does this proportion vary between the two sites? Again, it may be outside of the scope of this paper to measure this, but a discussion is warranted.

————————————

Grain size distributions: I had a difficult time understanding how the grain size distributions were characterized. Were the original distributions calculated using an area-by-number measurement and then transformed to a grid-by-number (or, equivalently, volume-by-weight) (e.g., following Bunte and Abt, 2001)? I'm just a bit confused about how the immobile fraction is transformed from 10% to 80% and 1% to 20% for the two study sites. Please be explicit about which distributions are being used, and perhaps consider using only the volume-by-weight equivalent to avoid confusion.

Also, the manuscript argues that the sediment sizes measured from aerial imagery are representative of the distribution of coarse sediment, but this is somewhat inconsistent with the rest of the analysis that argues that meter-scale boulders are at least partially mobile. This means that the deposits may also be winnowed with respect to boulders, and not just sediment finer than 30 cm. In other words, the river has likely moved more of the 1 m boulders than 2 m boulders since the boulders were deposited. This might be a small effect, though, especially if the mobile fraction is actually much finer than what is currently reported in the manuscript. ————————————

References not included in manuscript: Bunte, K., and S. R. Abt (2001), Sampling

surface and subsurface particle-size distributions in wadable gravel- and cobble-bed streams for analyses in sediment transport, hydraulics, and streambed monitoring, USDA Forest Service.

Ferguson, R. I. (2012), River channel slope, flow resistance, and gravel entrainment thresholds, Water Resour. Res., 48, W05517, doi:10.1029/ 2011WR010850.

Parker, C., N. J. Clifford, and C. R. Thorne (2011), Understanding the influence of slope on the threshold of coarse grain motion: Revisiting critical stream power, Geomorphology, 126(1–2), 51–65, doi:10.1016/j.geomorph.2010.10.027.

Prancevic, J. P., and M. P. Lamb (2015), Unraveling bed slope from relative roughness in initial sediment motion, J. Geophys. Res. Earth Surf., 120, doi:10.1002/2014JF003323.

Rickenmann, D., and A. Recking (2011), Evaluation of flow resistance in gravel-bed rivers through a large field data set, Water Resour. Res., 47, W07538, doi:10.1029/2010WR009793.
* * *

---

## Referee Comment (RC2) · Anonymous Referee #2 · 2 Jan 2019

Finnegan et al. present two case studies of valley blocking and boulder transport at two earthflows that occur in the Franciscan complex of California. They compare two sites, one near San Jose, where earthflows impinge on rivers with small upstream contributing areas and one on the Eel River, which occurs where the upstream contributing area is larger. The main difference between the two sites, is that boulders delivered by the earth flow block the valleys and generate considerable upstream aggradation at the San Jose site, but at the Eel River site, the channel is not blocked, despite evidence for the delivery of coarse boulders to the channel. The authors then assess thresholds of motion for boulders measured from satellite imagery and use stream gage data to compare the relative mobility of the boulders at the two sites. The conclusion that valley width influences blocking susceptibility has important implications for understanding how hillslope processes influence bedrock river incision, which has implications for how relative base-level fall is transmitted through catchments. In my opinion the manuscript makes a nice contribution and is appropriate for Earth Surface Dynamics. However, I have several comments regarding the analyses that should be addressed in a re-submission:

1. I agree with the comments by Referee Prancevic, who suggested a different approach for quantifying the threshold for motion. Costa (1983, GSA Bulletin) briefly summarizes how when assessing the motion of the largest particles in a channel, the relative roughness of the bed differs than for transporting the median bedload, etc., and such effects can be further considered in the manuscript.

2. Regarding the mobility of the boulders (e.g., (p. 11, line 24); if the boulders are mobile in a 2-yr recurrence flood, and presumably there have been many such floods (and larger floods) since the boulders were deposited, why are they still spatially co-located with the earthflow toes? It would be useful to provide a broader characterization of the spatial occurrence of boulders at each earthflow. For example, are they only present within the earthflow-influenced reach, or are they also located downstream as would be expected from fluvial transport?

3. I realize the scope of the manuscript is a case study comparison of the two sites, but it would be useful to further document blocking at small drainage area at other sites, as this is the main conclusion of the manuscript. Given the extensive earthflow observations generated by some of the co-authors for the Eel River watershed, it may be possible to assess blockage as a function of drainage area by either by inspection of available imagery or via measurements of floodplain width where suitable topographic data are available. A plot of the proportion of earthflows that block rivers as a function of drainage area could be informative, for example.

4. In the discussion of controls on blocking (p. 11), particle jams are noted as a possible

mechanism. It is difficult to discern from figures 13 and 14, but are the boulders actually touching one another? Whether they touch or not seems relevant to the arguments (e.g., force chains).

Editorial comments: P. 1 Line 17: replace "stream gages" with "stream gage data" P. 1 Line 18 and 20: replace "top" with "largest" P. 2 Line 9: year missing from citation (same citation in reference list is also missing co-authors) P. 3 Line 1: add a few words: . . .exploit "discharge data from" USGS. . . P. 3 Line 20 units are not consistent throughout; m/a here but m/yr elsewhere. P. 5 first paragraph: To be consistent with the rest of the text, keep the order the same, present Arroyo Hondo first, then the Eel River. P. 5 Line 20: Combine this sentence with the previous paragraph to avoid a one-sentence paragraph. P 8 Lines 9 and 13: USGS gage websites/numbers are already given, so this is repetative. P. 14 Line 14: extra ")" P. 14 Line 18: change "period" to "periodic"

---

## Author Comment (AC1) · 7 May 2019

We are grateful for the constructive comments of the two reviewers, which encouraged us to reshape the analysis in our resubmission. Below we address the key critiques of each reviewer separately and, where relevant, describe how we have addressed these critiques in our revised manuscript, which now has a new title:

"River Channel Width Controls Blocking by Slow-moving Landslides in California's Franciscan Mélange"

Before responding to the individual comments below, however, we wanted to provide a

[Figure]

brief overview of the key changes that the manuscript has undergone since the last submission. In the original submission, we derived a physically-based model to compute the drag force on a partially submerged boulder in order to determine the conditions under which it is mobile. Based in part on the reviewer comments, as well as on an informal review by Roman DiBiase, we realized that the boulder mobility analysis was too oversimplified to be useful for the problem at hand. Predicting the mobility of an isolated boulder on a smooth river bed (our first approach) is completely different than predicting the mobility of a boulder that is part of a boulder cascade, which is more typical of landslide deposited boulders. For this reason, we now simply use the Shields criterion to calculate the largest movable grain size in each river, which quickly leads to the conclusion that landslide derived boulders are very infrequently, or perhaps never, mobile in the rivers examined.

This realization, in turn, led us to explore another potential explanation for the clear difference in susceptibility to landslide blocking between the two river systems examined. We now argue that the dramatically different sensitivity of the two locations to landslide blocking is related to differences in channel width relative to typical seasonal displacements of earthflows. A synthesis of seasonal earthflow displacements in the Franciscan Mélange shows that the channel width of the Eel River is $\sim$ 5 times larger than the largest annual seasonal earthflow displacements. In contrast, during wet winters, earthflows are capable of crossing the entire channel width of Arroyo Hondo and Alameda Creek. Synthesis of boulder size distribution data, satellite imagery and hydraulic data suggests that in narrow channels earthflows can cross the channel and deposit channel spanning boulder jams that locally impede coarse sediment transport. In contrast, larger channels are able to flow around the toe of earthflows when they impinge on river channels, thereby preventing blocking. We emphasize that this effect is independent of the examined rivers' capacities to actually mobilize coarse debris.

RC1:

"Flow hydraulics and flow competence: My primary concern with the manuscript is the

line of reasoning used to calculate the maximum boulder size that can be transported by the Eel River and Arroyo Hondo."

See comments above. We have discarded the boulder mobility analysis in the revision.

"First, as far as I understand, the analysis relies on hydraulic measurements (flow depths, velocities, and discharges) that were made at USGS gage stations that are 2 km (for Oak Ridge earthflow) and 25 km (Mile 201 slide) downstream of the earthflow deposits. Flow depths, velocities, and discharges at the gage sites are not the same as those at the earthflow deposits. In particular, flow depth and velocity are both sensitive to changes in bed sediment size and channel width. The presence of coarse sediment (such as that found in the earthflow deposits) tends to reduce flow velocities and increase flow depths (e.g., Rickenmann and Recking, 2011). This phenomenon is particularly concerning for this analysis because it may reduce the tendency for boulders to emerge from the flow, which the manuscript proposes as the primary mechanism to stabilize boulders."

This is a good point. While we have discarded the boulder mobility calculation presented in the first submission, we still perform a Shields stress calculation in the new submission and use it to assess boulder mobility. Hence, it is worth considering how the deposition of landslide debris changes coarse sediment transport capacity. That said, given our results that landslide-derived boulder mobility seems very unlikely in the settings examined here, we decided not to embark on a modeling effort to explore the morphodynamics of boulder cascades associated with landslides. However, we now note explicitly that our coarse sediment mobility calculations (page 9, lines 16-22):

'ignore the possible morphodynamic feedbacks that might result from the deposition of large boulders in a channel. On the one hand, landslide derived boulder deposits are steep relative to points upstream and downstream (Figures 5A, 6A), suggesting that the deposition of landslide debris might lead to conditions more favorable to coarse sediment transport. On the other hand, large boulders exert substantial drag on the

flow, which can completely offset increases in coarse sediment transport capacity due to the steeper slopes of boulder cascades (Schneider et al., 2016). For this reason, we simply consider the coarse sediment transport capacity of the river at the gage sites as an index of the river's ability to move coarse landslide-derived debris independent of changes in bed morphology caused by that debris.'

"Second, I am not sure that it is necessary to introduce a new, untested calculation for boulder mobility when tested models already exist."

We agree. Again, see comments above.

"Simultaneously accounting for the effect of boulders on sediment mobility and flow resistance (for calculation of flow velocity and depth) is tricky and prone to error, but these difficulties can be circumvented by using a critical dimensionless stream power to calculate flow competence (Parker et al., 2011). This criterion has been shown to be a more reliable predictor of sediment mobility than the critical Shields stress in shallow, rough flows (Parker et al., 2011; Ferguson, 2012; Prancevic and Lamb, 2015), and it only requires information about discharge, width, slope, and grain size, all of which the authors have measured at the earthflow deposits.

In a quick calculation, I found that this method predicts that the largest mobile sediment during the bankfull flood is âĹij20 cm for both sites. I may have missed something, but this would suggest that there is not a large difference in flow competence between the two sites. Moreover, these grain sizes seem much more reasonable to me than 2.4 m and 4.9 m. (Can a 4 m boulder really be transported by a 2-year flood in the Eel?) This estimate of the mobile size fraction is also consistent with the observation that there are very few boulders larger than 30 cm found in the river outside of earthflow deposits at both sites."

We agree with this sentiment. We now use a Shields stress calculation (Table 1) in the draft, which yields very similar results to the back of the envelope calculation performed by RC1: 22 cm 2-year mobility threshold on the Eel River, 31 cm 2-year mobility threshold on Arroyo Hondo. Because our focus is on the mobility of the river independent of the changes that landslides cause to the river (which is an interesting topic, but beyond the scope of what we are trying to accomplish here), we are comfortable using the Shields-based approach, which is not rooted specifically in modeling boulders. For this reason, and because our results are now comparable to RC1's back of the envelope calculation, we have not changed the approach in the paper to that suggested by RC1.

"Comparison of longitudinal profiles: I do think that the longitudinal profiles (Figs. 5-7) show different earthflow signatures between the study sites, but I think that the comparison would be more compelling if it were more even. The profile of the Eel River is 3x to 4x longer than the other two profiles but is squeezed into the same plot size, which makes it difficult to compare the topographic imprint of earthflows (which seem to not change substantially in size between the two study areas)."

Actually, together the Arroyo Hondo and Alameda Creek sites represent 19 km of river distance and the Eel River site is 30 km, so they are comparable in size.

"Moreover, the slope measurements on the Eel River were made at a resolution 10x larger than at the other study sites, potentially smoothing over some of the variability in slope. I think it would be more convincing to use the same spacing for slope measurements and to zoom in on a portion of the Eel River profile such that the profile examined a similar length to the others two."

This is a good point. To deal with this problem, we have actually done away completely with measuring channel slope in the new draft. Instead, we linearly detrend the river profiles and then plot the residual topography (Figures 5b,6b,7b). The amplitude of the residual topography provides a nice means of identifying perturbations in the channel long profile induced by landslides that are free of the smoothing issues pointing out by RC1. We describe this in more detail in the methods.

"Boulder supply rate: The manuscript focuses on the relative size distributions of coarse sediment supplied to the river between the two study areas. However, it may

also be more important to consider the supply rate of coarse boulders. If boulders are delivered very slowly, then the river can rely on bigger, rarer flood events to remove the boulders. It may be outside of the scope of this paper to consider this effect quantitatively, but it should at least be discussed, especially because the Oak Ridge earthflow seems to be sliding 5x faster than the Mile 201 slide. This could be part of the explanation for why there are more big boulders in the river next to the Oak Ridge earthflow.

Also, if the supply rate is important, it is not only the distribution of coarse sediment delivered by the earthflow that matters, but also the proportion that is coarse sediment. What portion of earthflow-derived material is just fines that is just being easily washed away, and does this proportion vary between the two sites? Again, it may be outside of the scope of this paper to measure this, but a discussion is warranted"

This is a good point. To address this issue in the revision we provided a new analysis in which we compared the volumetric flux per unit river channel width for the Boulder Creek earthflow (the largest earthflow along the Eel River) and for Oak Ridge earthflow. This new section (Page 10, lines 18-34) shows that 'although the Boulder Creek earthflow has an order of magnitude larger volumetric flux ($\sim$ 15,000 mˆ3/yr) than Oak Ridge earthflow (1700 mˆ3/yr), the Eel River has an order of magnitude larger channel width (125 m) than Arroyo Hondo (12 m). Hence, earthflow fluxes per unit channel width at the two sites are nearly identical, $\sim$140 mˆ3/m for Arroyo Hondo and $\sim$130 mˆ3/m for the Eel River. Despite this similarity, there is no evidence of blocking in the long profile of the Eel River at the location of the Boulder Creek slide, whereas the channel of Arroyo Hondo is clearly blocked at Oak Ridge (Figure 5). For this reason, we also rule out boulder supply relative to transport capacity as a likely driver of the observed morphological differences on the two rivers.'

"Grain size distributions: I had a difficult time understanding how the grain size distributions were characterized. Were the original distributions calculated using an area-by-number measurement and then transformed to a grid-by-number (or, equivalently,

volume-by-weight) (e.g., following Bunte and Abt, 2001)? I'm just a bit confused about how the immobile fraction is transformed from 10% to 80% and 1% to 20% for the two study sites. Please be explicit about which distributions are being used, and perhaps consider using only the volume-by-weight equivalent to avoid confusion."

In the revision, we no longer include the conversion to volume as this is no longer relevant given the change in our findings regarding boulder mobility.

"Also, the manuscript argues that the sediment sizes measured from aerial imagery are representative of the distribution of coarse sediment, but this is somewhat inconsistent with the rest of the analysis that argues that meter-scale boulders are at least partially mobile. This means that the deposits may also be winnowed with respect to boulders, and not just sediment finer than 30 cm. In other words, the river has likely moved more of the 1 m boulders than 2 m boulders since the boulders were deposited. This might be a small effect, though, especially if the mobile fraction is actually much finer than what is currently reported in the manuscript."

As suspected by RC1, our results now show that the material capable of being transported in the two rivers is much smaller than what we indicated in the first draft. Hence, we anticipate that the effect described above is probably not fundamental to our analysis.

RC2:

"1. I agree with the comments by Referee Prancevic, who suggested a different approach for quantifying the threshold for motion. Costa (1983, GSA Bulletin) briefly summarizes how when assessing the motion of the largest particles in a channel, the relative roughness of the bed differs than for transporting the median bedload, etc., and such effects can be further considered in the manuscript."

We believe this issue has now been resolved given the changes that we have implemented in the submission, as described at the start of this document, as well as in our
response to RC2:

While we have discarded the boulder mobility calculation presented in the first submission, we still perform a Shields stress calculation in the new submission and use it to assess boulder mobility. Hence, it is worth considering how the deposition of landslide debris changes coarse sediment transport capacity. That said, given our results that landslide-derived boulder mobility seems very unlikely in the settings examined here, we decided not to embark on a modeling effort to explore the morphodynamics of boulder cascades associated with landslides. However, we now note explicitly that our coarse sediment mobility calculations (page 9, lines 16-22):

'ignore the possible morphodynamic feedbacks that might result from the deposition of large boulders in a channel. On the one hand, landslide derived boulder deposits are steep relative to points upstream and downstream (Figures 5A, 6A), suggesting that the deposition of landslide debris might lead to conditions more favorable to coarse sediment transport. On the other hand, large boulders exert substantial drag on the flow, which can completely offset increases in coarse sediment transport capacity due to the steeper slopes of boulder cascades (Schneider et al., 2016). For this reason, we simply consider the coarse sediment transport capacity of the river at the gage sites as an index of the river's ability to move coarse landslide-derived debris independent of changes in bed morphology caused by that debris.'

"2. Regarding the mobility of the boulders (e.g., (p. 11, line 24); if the boulders are mobile in a 2-yr recurrence flood, and presumably there have been many such floods (and larger floods) since the boulders were deposited, why are they still spatially co-located with the earthflow toes? It would be useful to provide a broader characterization of the spatial occurrence of boulders at each earthflow. For example, are they only present within the earthflow-influenced reach, or are they also located downstream as would be expected from fluvial transport?"

This is an excellent point. Given the results of our updated mobility calculation, the

**ESurfD**

Interactive
comment

observation that boulders are clustered at the toes of earthflows and not downstream makes sense. In response to this comment, in the revision we now argue that 'it's entirely plausible that the entire distribution of boulder sizes delivered by earthflows is immobile once delivered to channels in both locations. This interpretation is supported by the fact that gravel bars downstream of the two reference earthflow sites do not contain boulders, and typically do not contain clasts that are even discernable above the $\sim$ 30 cm resolution of the imagery.'

3. I realize the scope of the manuscript is a case study comparison of the two sites, but it would be useful to further document blocking at small drainage area at other sites, as this is the main conclusion of the manuscript. Given the extensive earthflow observations generated by some of the co-authors for the Eel River watershed, it may be possible to assess blockage as a function of drainage area by either by inspection of available imagery or via measurements of floodplain width where suitable topographic data are available. A plot of the proportion of earthflows that block rivers as a function of drainage area could be informative, for example.

We agree with the sentiment of this comment, but feel that such an undertaking is better suited for a future study. We feel that the strength of this contribution is the detailed comparison of the two sites and prefer not to bring in a more synoptic but less detailed analysis in this paper.

"In the discussion of controls on blocking (p. 11), particle jams are noted as a possible mechanism. It is difficult to discern from figures 13 and 14, but are the boulders actually touching one another? Whether they touch or not seems relevant to the arguments (e.g., force chains)."

We have improved these figures (now 12 and 13), and in particular removed the boxes around each boulder that were present in the first submission. We now hope that it is much easier to see the clusters of boulders in each channel and how they are in fact touching, as is clear in the field.

[Figure]

Editorial comments:

P. 1 Line 17: replace "stream gages" with "stream gage data" P. 1 Line 18 and 20: replace "top" with "largest"

These edits are no longer relevant as the section in question has been re-written.

P. 2 Line 9: year missing from citation (same citation in reference list is also missing co-authors)

Fixed. P. 3 Line 1: add a few words:. . .exploit "discharge data from" USGS. . .

This edit is no longer relevant as the section in question has been re-written.

P. 3 Line 20 units are not consistent throughout; m/a here but m/yr elsewhere.

Changed to m/yr

P. 5 first paragraph: To be consistent with the rest of the text, keep the order the same, present Arroyo Hondo first, then the Eel River.

This edit is no longer relevant as the section in question has been re-written.

P. 5 Line 20: Combine this sentence with the previous paragraph to avoid a one-sentence paragraph.

Done

P 8 Lines 9 and 13: USGS gage websites/numbers are already given, so this is repetative.

We deleted the redundant reference in the revision.

P. 14 Line 14: extra ")"

Changed

P. 14 Line 18: change "period" to "periodic

This edit is no longer relevant as the section in question has been re-written.

---

## Author Response (AR2)

**Comments in Black**

*Responses in Italics*

Referee #1:

**I again enjoyed reading the revised version of this manuscript. The authors did an excellent job of addressing the previous set of reviewer comments, and I find the revised story to be much more compelling. I have only one (very) small remaining concern about the interpretation of the evidence presented in the manuscript, and that can likely be easily addressed. My concern is that the authors use earthflow flux as a direct measurement of boulder supply rate, when comparing the boulder supply rates of the two earthflows (Boulder Creek and Oak Ridge). However, this assumes that the two earthflows contain the same fraction of boulders. Isn't it possible that one earthflow contains 10% boulders, while the other contains 50% boulders? This assumption should be stated explicitly in the analysis, perhaps with some anecdotal justification from the authors' experience walking around these two landscapes. Other than that, I have just a few suggestions that require no response.**

*This is a good point that we neglected to address in the previous revision. We have now added a sentence (10.31-32) that states: "Assuming a similar concentration of boulders within the mélange at both sites, which is reasonable based on the surface distribution of boulders that is apparent at both sides, this calculation suggests that boulder fluxes per unit channel width at the two sites are also likely to be comparable."*

**Line comments (no response required):**
**line 7.9.        missing period**

*Fixed*

**lines 8.15 - 17. Why not show the bankfull flow depth on Figs. 5C, 6C, and 7C?**

*River width, although much less variable than valley width, is not completely constant.  We felt that the addition of another noisy curve on these figures would make them more difficult to read.*

**Sec. 3.2. But this does not mean that the boulder concentration is the same, because we don't know the what proportion of the earthflow material is boulders for either earthflow. They could potentially be very different. Perhaps you could draw on your collective field observations to make the case that boulder fractions are similar between to the two sites?**

*See comments above*

**line 9.29. Is "Highway 201" supposed to be "Mile 201?"**

*Yes, thanks. Fixed.*

**lines 10.30 - 31.      How did time fall out of the units for the fluxes? Should be m^2/yr?**

*Thanks for catching this. This is fixed in the new draft.*

**lines 10.34 - 35.      I agree that this is probably the case, but it's not clear that boulder supply rates per unit channel width are the same between the two sites just because earthflow fluxes are the same between the two sites. See comment above on Sec. 3.2.**

*See comments above*

**line 13.7      Missing space "2016)generally"**

*Fixed*

**Fig. 7. Does this figure look way too busy if the earthflow reaches are shaded in red, similar to Figs. 5 and 6?**

*We added red shading to figure 7. It's busy, but perhaps better to be consistent with the other two figures.*

Referee #2:

**I have carefully read the revised manuscript, as well as the author responses to the comments from both reviewers. I find that the authors have substantially improved the analysis and have produced a clear manuscript that contributes to the understanding of the interaction of earth flows and rivers and proposes new testable hypotheses.**

**Minor editorial comments:**
**page 4, Line 34: The references to the figures is out of sequence, e.g., Fig. 5b is cited before Fig. 5a. Given that this section describes the methods, a reference to the figures is probably not needed until the Results section.**

*We removed the references to the figures in this location.*

**page 13, Line 7, insert a space before "generally"**

*Fixed*

**Also, capitalization in figure names is not consistent (e.g., Fig. 5A, Fig. 5b)**

*Resolved*

**The discussion of Schumm and Stevens (1973-Geology), is of relevance to the final line of the manuscript regarding size reduction of boulders.**

*Thanks. This has been added.*

[revised manuscript text omitted]